# Influence of tectonics on global scale distribution of geological methane emissions

Giancarlo Ciotoli[1,2], Monia Procesi[2 ✉], Giuseppe Etiope[2,3], Umberto Fracassi[2] & Guido Ventura[2]

Earth's hydrocarbon degassing through gas-oil seeps, mud volcanoes and diffuse micro-seepage is a major natural source of methane ($CH_4$) to the atmosphere. While carbon dioxide degassing is typically associated with extensional tectonics, volcanoes, and geothermal areas, $CH_4$ seepage mostly occurs in petroleum-bearing sedimentary basins, but the role of tectonics in degassing is known only for some case studies at local scale. Here, we perform a global scale geospatial analysis to assess how the presence of hydrocarbon fields, basin geodynamics and the type of faults control $CH_4$ seepage. Combining georeferenced data of global inventories of onshore seeps, faults, sedimentary basins, petroleum fields and heat flow, we find that hydrocarbon seeps prevail in petroleum fields within convergent basins with heat flow $\leq 98\,mW\,m^{-2}$, and along any type of brittle tectonic structure, mostly in reverse fault settings. Areas potentially hosting additional seeps and microseepage are identified through a global seepage favourability model.

[1] Consiglio Nazionale delle Ricerche, Istituto di Geologia Ambientale e Geoingegneria, Via Salaria km 29300, 00015 Monterotondo, Rome, Italy. [2] Istituto Nazionale di Geofisica e Vulcanologia, Via di Vigna Murata 605, 00143 Rome, Italy. [3] Faculty of Environmental Science and Engineering, Babes-Bolyai University, Str. Fantanele 30, Cluj-Napoca, Romania. ✉email: monia.procesi@ingv.it

Among the natural sources of greenhouse gases that may contribute to climate changes, geological emissions of carbon dioxide ($CO_2$) and methane ($CH_4$) from Earth degassing have a specific role, as evidenced by field measurements, inventories, process-based models, bottom-up and top-down estimates[1–6]. $CO_2$ degassing from volcanic and geothermal areas may have played a climatic role over geological-time scales[7,8] but it appears to represent a minimal component in the present-day global $CO_2$ atmospheric budget; global geo-$CO_2$ emission (likely < 1000 Mt yr$^{-1}$)[1,4] is three orders of magnitude lower than the total $CO_2$ emissions from natural plus anthropogenic sources[9]. Quite another matter is the relative importance of geological $CH_4$ (hereafter geo-$CH_4$) emissions: bottom-up and top-down estimates suggest that geo-$CH_4$ emission globally amount to ~45 Mt yr$^{-1}$[6,10], about 8% of total (natural plus anthropogenic) $CH_4$ sources (~560 Mt yr$^{-1}$)[11]. Lower geo-$CH_4$ emissions derived from preindustrial-era ice core radiocarbon $^{14}CH_4$ analyses[12] opened a debate, suggesting the need of further checks of global geo-$CH_4$ estimates. In this respect, an important step is improving the knowledge of global seepage distribution and extension, and its controlling geological factors. While geo-$CO_2$ degassing is mainly controlled by extensional tectonics, along normal faults and rift systems that mostly drive $CO_2$ release from crustal geothermal reservoirs, magmatic chambers and the mantle[5,7,8], geo-$CH_4$ degassing (seepage) takes place primarily in sedimentary, petroleum (oil and gas)-rich basins, where $CH_4$ can have a microbial or thermogenic origin[2,6,13–15]. However, the role of the basin geodynamics (convergent or divergent) and the type of faulting (reverse, normal, strike-slip) on gas seepage was examined only sporadically, without statistical quantification on a global scale[14,16]. $CH_4$ migration and seepage are primarily driven by advection (Darcy's flow), controlled by gas pressure and rock permeability related to fracture networks and faults[15]. Whether a specific type of tectonics and fault (such as extensional tectonics for $CO_2$) is needed for $CH_4$ degassing is unknown. Addressing this issue is critical to assess the pathways of $CH_4$ release and to identify potential, not yet inventoried, geo-$CH_4$ emission regions, including those hosting the invisible, diffuse exhalations (microseepage)[6,17,18]. Knowing the area and spatial distribution of methane seepage is fundamental for both atmospheric methane budget studies and petroleum exploration. In the first case, it allows to refine global and regional estimates of geo-$CH_4$ emission to the atmosphere: the area where the emission takes place represents in fact the activity to be associated to the average gas flux, over that area (emission factors), in bottom-up gas emission estimates[6,10,11]. This is particularly critical for microseepage, which represents a major class of geo-$CH_4$ source, estimated in the order of 15–33 Mt yr$^{-1}$ [10]. In the second case, maps of spatial distribution of seepage in sedimentary basins may drive the exploration for the identification of subsurface petroleum reservoirs[14]. Knowing the type of tectonics favouring hydrocarbon seepage is also important in the study of potential methane source regions on other planets as Mars, where recent atmospheric $CH_4$ detections have raised the question on what are the tectonic features that may have released the gas[19].

Here, we address the above issues using geospatial analysis of global datasets of onshore hydrocarbon seeps (Supplementary Notes 1 and 2 and Supplementary Fig. 1). We analyse the different geological factors that could control the $CH_4$ seepage on continents, e.g., the existence of petroleum fields, type of sedimentary basins (convergent and divergent), heat flow and fault type. The work does not include offshore seeps because available inventories, referring to relatively wide areas, do not report geographic coordinates of individual seeps[6]. The role of tectonics is investigated by fault density maps and by examining the spatial association between seeps and type of faulting (a logical flow scheme is shown in Supplementary Fig. 2). We show that geo-$CH_4$ seepage preferably develops in convergent basins, secondarily in divergent settings, and is mainly associated with reverse faults. Gas seepage however occurs along any type of brittle tectonic structure. We develop a model of seepage favourability with the aim to identify potential seepage areas not documented so far.

## Results

**Influence of basin geodynamics and petroleum fields**. We applied GIS-based geospatial and geostatistical analyses to evaluate the relationship between hydrocarbon seep distribution, basin geodynamics and petroleum fields (see Methods).

Convergent basins include retroarc, forearc, arc-related wrench and foreland basins associated with fold-and-thrust belts and form in geodynamic settings characterised by continental shortening and tectonic loading[20] (Fig.1a). Divergent basins include intra-cratonic, rift and post-rift sags, passive margins and wrenches (see details in Supplementary Note 3).

Geospatial analysis shows that out of 2699 seeps, 1941 seeps (72%) occur in convergent basins (Table 1 and Supplementary Fig. 3), although these basins cover an area smaller than that of divergent basins (Fig. 1b and Table 1).

There is not a well defined prevailing type of seep in convergent basins: 33% are gas seeps, 30% oil seeps and 37% mud volcanoes. About 95% of the worldwide mud volcanoes occur in these basins (Fig. 1b), which are closely related to thrust systems and sedimentary diapirism[21]. Divergent basins host 758 seeps (28% of the total), of which 70% is oil, 26% gas and 4% mud volcanoes (Fig. 1b).

Convergent and divergent basins host 58% and 42% of petroleum fields, respectively. As expected, most seeps (81%) occur within the area covered by petroleum fields (Fig. 2a; Table 1). The remaining 19% of seeps can be related to minor fields not included in the petroleum field inventory, direct fluid migration from source rocks, also not included in the inventory, and long-distance lateral migration of gas and oil[14]. Seeps prevail at the margins of petroleum fields, which are typically more faulted and fractured (examples in Supplementary Note 3 and Supplementary Fig. 4). The result of our analysis demonstrates that seeps are a fundamental component of petroleum systems according to the definition of Petroleum Seepage System[22], and gas-oil reservoirs are the source of most seeps[15]. In addition, seepage mostly (95%) occurs in areas with heat flow ≤98 mWm$^{-2}$ (Supplementary Table 1), which are values typical of the thermal status of petroleum systems[23] (Supplementary Note 3 and Supplementary Fig. 5).

**Seeps and faults**. It is known that hydrocarbon seeps, especially gas seeps and mud volcanoes, develop along fracture and/or fault networks[14,24]. Therefore, we have evaluated the spatial occurrence of faults within convergent and divergent basins and within the petroleum fields. We built a fault density map (1° × 1°) as a proxy of brittle tectonics occurrence using onshore, active and non-active faults from a worldwide fault database integrating published global and regional datasets (Supplementary Figs. 6 and 7a–e). Fault density, in terms of number of faults, weighted for their length (km) per km$^2$ ($N_f$ km$_f$ km$^{-2}$) is mapped by using a kernel density algorithm[25,26] (Methods, Supplementary Note 4 and Supplementary Figs. 8 and 9a–c). We then compared the statistical distribution of the global fault density with fault density values in convergent and divergent basins in petroleum fields and at seep locations (Fig. 2b). Results show that convergent basins are, on average, more fractured than divergent basins (mean fault density values are 274.7 and 102.9 $N_f$ km$_f$ km$^{-2}$, respectively). This is likely due to the presence, in convergent basins, of longer

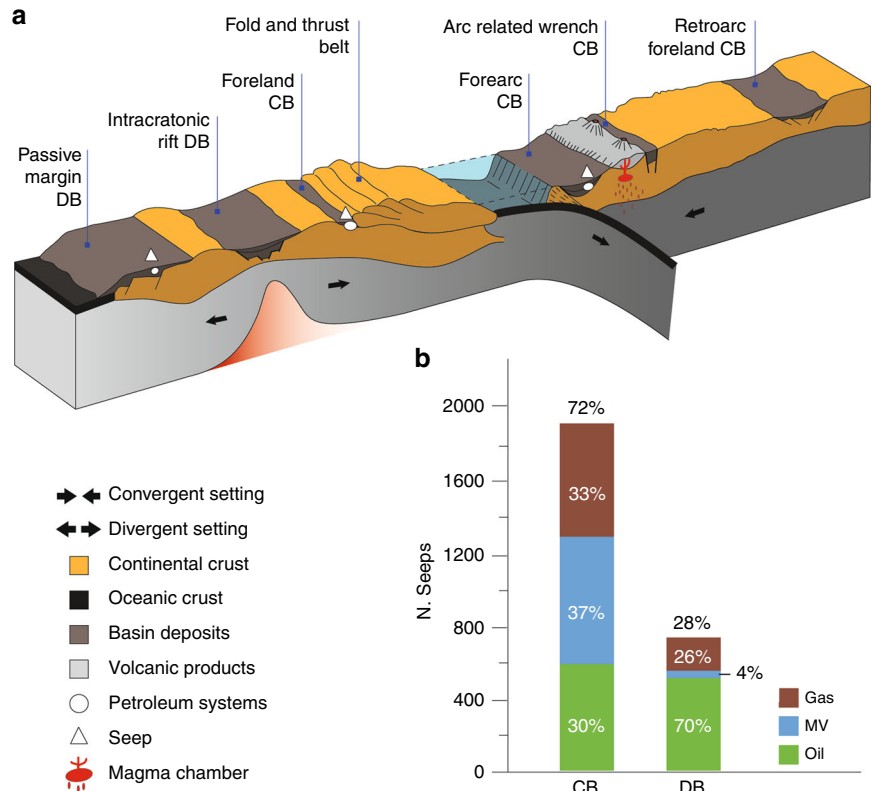

**Fig. 1 Simplified conceptual geological model of convergent and divergent geodynamic settings. a** The main basin typologies and the potential occurrence of petroleum systems and seeps are also indicated. **b** Bar plot of seeps distribution: gas seeps, mud volcanoes (MV) and oil seeps, in convergent (CB) and divergent (DB) basins.

**Table 1 Seep distribution in sedimentary basins and petroleum fields.**

|  | Area (Mkm²) | N° of seeps |  | % seeps | N° of petroleum fields | % petroleum fields |
|---|---|---|---|---|---|---|
| Convergent basins | 44.5 | 1941 | gas 644 oil 589 MV 708 | 72 | 517 | 58.0 |
| Divergent basins | 73.8 | 758 | gas 195 oil 530 MV 33 | 28 | 374 | 42.0 |
| Petroleum fields | 22.7 | 2179 |  | 81 | – | – |

Number (and percentage) of seeps within convergent and divergent basins and petroleum fields, and number (and percentage) of petroleum fields within the two groups of basins.
*MV* mud volcanoes.

faults, mainly thrusts associated with orogenic belts. These fault systems concentrate along the edges of thrust-and-fold belts and may be related to ruptures from shallow decollement layers up to the surface with multiple short-cuts, back-thrusts, and ancillary high-angle fracturing[20,27]. Petroleum fields show a mean fault density value (202.7 $N_f$ km$_f$ km$^{-2}$) comparable to that of convergent basins. Interestingly, seeps occur for any value of fault density. This means that both isolated faults and regions with fault clusters can host seeps (Fig. 2b). Petroleum fields and seeps prevail in convergent basins (Table 1). This can be explained by the fact that convergent basins, although covering a smaller area than that of divergent basins, are more fractured and faulted (Fig. 2b)[28].

To summarise, we found that seeps occur for any type of fault density, preferably at the boundaries of petroleum fields hosted mainly in convergent basins with heat flow values ≤98 mWm$^{-2}$ (Supplementary Note 3).

We evaluated the role of faulting regime in gas seepage by correlating the seep location and type to the style (reverse, normal, and strike-slip) of the fault.

The distance between each seep and the nearest fault is calculated by proximity analysis (see Methods, Supplementary Note 4 and Supplementary Table 2). We stress that the nearest fault considered in this analysis solely serves as the source of information on the faulting regime of the area associated with seepage, and does not necessarily represent the actual fault system along which fluids migrated originating the seep.

The results show that the distance between each seep and the nearest fault, used to extract information on the fault style, is mostly less than 20 km (Fig. 3a, Supplementary Fig. 10). This value corresponds to the nearest threshold distance common to all types of seeps (Supplementary Fig. 11) and it is within the uncertainty of geographic position of either seep and/or global fault inventories (see Methods). Accordingly, this distance has

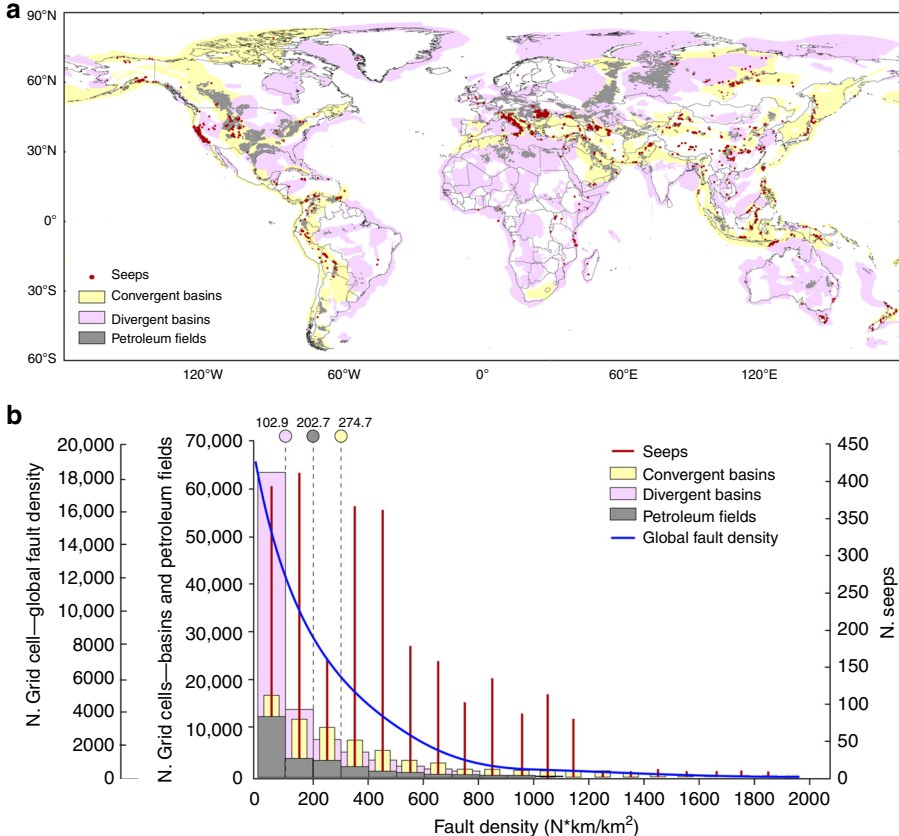

**Fig. 2 World map of sedimentary basins, petroleum fields and seeps, and related fault density. a** Map of sedimentary basins (convergent, light yellow; divergent, light violet), petroleum field areas (grey) and geo-CH$_4$ seeps (red dots). **b** Multiple histograms of fault density values at global scale (blue line), in convergent (light yellow) and divergent (light violet) basins, in petroleum fields (grey), and at seep location (red). Grid cell is 0.2° × 0.2°. The fault density mean value for divergent basins, petroleum fields and convergent basins is indicated above the light violet, grey and yellow circles, respectively.

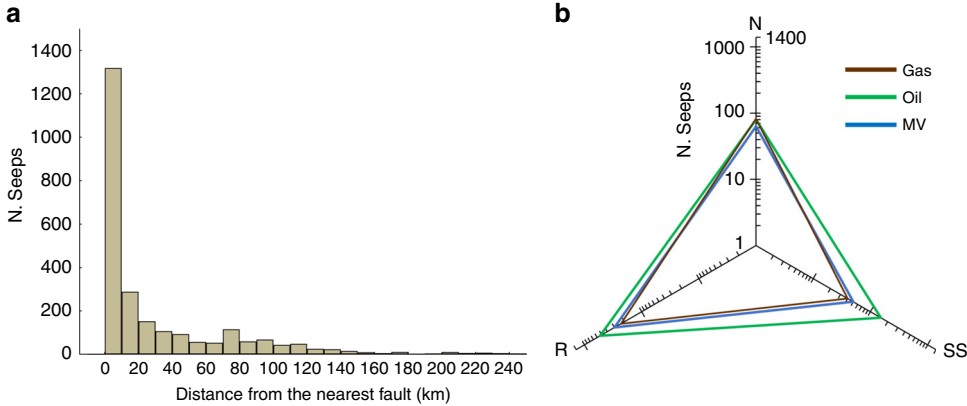

**Fig. 3 Seeps versus distance from faults and fault type. a** Histogram of the distances between seeps and the nearest fault bearing the faulting style information, as available in the global fault dataset. The distinction by fault type is shown in Supplementary Fig. 10. **b** Radar diagram showing the number and the type of seeps related to fault type (nearest fault); N: normal, SS: strike-slip, R: reverse; Gas: gas seeps, Oil: oil seeps, MV: mud volcanoes.

been used to analyse the association between seeps and fault type. The radar diagram in Fig. 3b shows that all types of seeps, i.e., gas seeps, oil seeps and mud volcanoes, occur in areas characterised by any fault style, with a preference for reverse faults, which is the type of faulting more frequent within petroleum field areas (Supplementary Table 3). The close spatial relation between mud volcanoes and reverse faults was evidenced in local and regional studies and it can be related to low-angle thrusts[14,16]. Also, mud volcanoes can be observed in strike-slip/transpressive[29–31], and extensional[32,33] settings. Extensional basins are characterised by higher rates of sedimentation, a depositional condition necessary for the trapping and migration of hydrocarbons[14]. A case study in Northern Italy demonstrates that mud volcanoes tend to occur on top of thrust-related anticlines hosting the main reservoir[34]. Our global analysis reveals that oil seeps are more frequently associated with reverse faults. It must be noted, however, that oil seeps may be not associated with any fault, as oil can migrate along permeable stratigraphic layers (e.g., homocline seeps)[15,24]. In our database, for example, we recognise 134 oil seeps located at distances >100 km from the nearest fault (most of them in the

cratonic Siberian oil provinces). The association between gas seeps and normal faults (Fig. 3b) is similar to the one observed for geothermal $CO_2$ degassing, which is mainly controlled by extensional structures[5]. This is the only analogy between $CH_4$ and $CO_2$ degassing and fault association. Basically, our results show that hydrocarbon seepage is not exclusively associated with any given fault type. The observed association between the tectonic style and seepage can, in theory, be also applied to offshore areas, where presently geospatial analyses cannot be performed due to the lack of a global inventory with precise seep geographic location[6].

**Global spatial model of geo-$CH_4$ seepage favourability**. On the basis of the results of the geospatial analysis, we developed a conceptual spatial model of seepage occurrence providing a global map of the potential geo-$CH_4$ emission areas. We used a GIS-based Spatial Multi-Criteria Decision Analysis (SMCDA) and, in particular, the Analytical Hierarchy Process (AHP) (Methods and Supplementary Note 5). The conceptual spatial model of seepage occurrence includes the following geological input parameters: basin type (convergent, divergent); global petroleum fields; heat flow; fault density, and faulting style. These factors are weighted by using AHP, an effective tool for determining the best combinations of factors. The model, represented by a raster map with $1° × 1°$ grid cell, has been normalised to obtain a final favourability map of seepage, expressed as spatial probability (%) of occurrence (Fig. 4a and Supplementary Fig. 12).

The model has been validated by verifying that the number of known (inventoried) seeps increases with seepage favourability. The test shows that about 70% of the seeps fall in the probability exceeding 50% (Fig. 4b). The model also suggests that, in addition to the known seepage areas (black dots in Fig. 4a), wide areas of geo-$CH_4$ emissions potentially occur in North America, Northern and Arctic Europe, Western Russia, Caucasus and Eastern Europe, Western Siberia, China, Turkmenistan, Kyrgyzstan and the Arabian Peninsula (Supplementary Note 5, Supplementary Fig. 12). The favourability model also allows to identify potential areas of microseepage, i.e., diffuse degassing of methane and other light hydrocarbons. The favourability model is in fact applicable to any type of seepage, focused (seeps or macro-seeps) and diffused (microseepage), since the gas migration mechanism (fundamentally advective, i.e., driven by pressure gradients, and whose intensity is mainly controlled by fault-related permeability) is the

same[15,18,22]. Besides the relevance for oil-gas exploration[14,35], microseepage was estimated as the major geo-$CH_4$ source to the atmosphere (about 24 (15–33) Mt yr$^{-1}$)[6,10]. The main uncertainty in the global $CH_4$ source strength is related to the limited knowledge of the global microseepage area. Our model suggests a high probability (≥50%) microseepage area of 8.1 Mkm$^2$, a value consistent with that previously predicted by process-based modelling (8.6 Mkm$^2$)[6].

## Discussion

As mentioned above, the results of our global geospatial analysis are consistent with previous qualitative and local scale observations[14,16]. We also checked specific examples where the type of fault in correspondence with the seepage is known. For example, gas seepage along normal faults occurs in the Katakolo Bay in Greece[36] and in the Tiber Delta in Italy[37]. The Giswil seep (Switzerland) is an example of gas exhalation along a strike-slip fault[38]. Mud volcanoes along thrust faults were studied in particular in Northern Italy and Azerbaijan[16,21].

We have shown that the tectonics style does not significantly affect the methane release. In fact, while compressional and extensional stresses may play a role in controlling water and oil migration (i.e., hydraulic conductivity), they do not significantly affect the gas-bearing property of faults, as a fault that is impermeable to water can be permeable to gas, as well known in studies on gas flow in fractured media[39–41]. Other factors, including lithology, rheology, fault activity and self-sealing processes (e.g., secondary mineral depositional processes) control the gas permeability of a fault[42,43].

The role of fault activity could not be examined in our geospatial analyses, because available datasets include only active faults (GEM dataset) or do not distinguish active from inactive faults. However, gas migration studies demonstrated that non-active, ancient faults in stable and poorly fractured terrains (e.g., in granites, within cratons) can also be gas-bearing[44,45]. This is consistent with our analysis that showed no relationship between fault density and seepage.

The fact that $CH_4$ can potentially be released by any type of fault is a key conclusion in fixing the criteria to search areas of $CH_4$ release on Mars, where reverse and normal faults are recognised[46]. Recent $CH_4$ spike detections in the Martian atmosphere have raised the question about the tectonic structures potentially favouring the release of methane[19]. By analogy with

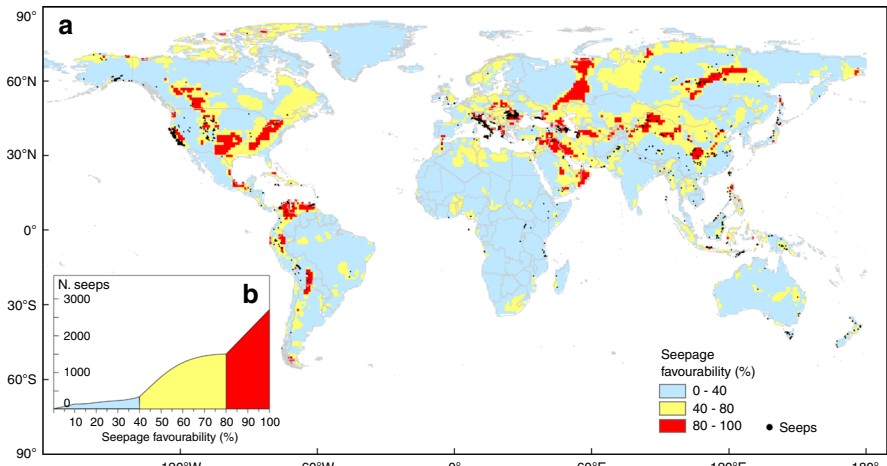

**Fig. 4 Favourability map of gas seepage. a** Map with the boundary of seepage probability classes selected according to the slope changes in the inset graph. The map suggests the potential areas where additional seeps (not identified in available inventories) and microseepage can occur. **b** Cumulative number of seeps.

what is observed on Earth, any type of fault on Mars, regardless the fault density of the area, may potentially be methane-bearing. Therefore, even isolated faults, either the extensional faults of the Tharsis region and along the Martian dichotomy, or the compressional faults in the lowlands (e.g., Acidalia, Utopia) and in the highlands (e.g., Arabia Terra) can potentially host gas seepage. Like on Earth, potential mud volcanism on Mars should preferentially occur within reverse fault areas. Morphological structures resembling mud volcanoes were actually identified in regions with compressional tectonic settings[47–49]. Whether methane source rocks, which are essential for Earth's seepage, do exist on Mars as yet remains an unknown factor[49].

## Methods

**Datasets.** Reported input data refer to a series of geological and geographic databases available in shapefile (either points, polylines or polygons), csv, ASCII, and/or grid formats, managed and elaborated in ArcGIS Pro (Copyright © 1999-2018 Esri Inc.). Geographic coordinates of all datasets are expressed in decimal longitude and latitude in the GCS WGS 1984 world projection system. All raster maps are reported in 1° × 1° square grid format. The source of the several datasets (onshore seeps, sedimentary basins, petroleum fields, heat flow and faults) are described in Data Availability.

**Geospatial analysis.** All datasets were analysed by using geospatial and geostatistical analyses in GIS environment. The output raster values of 1° × 1° grid maps were extracted at each seep location using the Spatial Analyst Tool (Extract Multi Values to Points command).

The Kernel Density algorithm was used to calculate the fault density map, weighted by fault length, expressed as number of faults multiplied by fault length and divided by unit area ($N_f \cdot km_f \, km^{-2}$) (Supplementary Figs. 8 and 9a–c). The kernel function is described in Silverman[25]. We calculated the density of faults around each output raster cell by fitting a smoothly curved surface over each fault line. The surface value is higher at the location of the fault and decreases with increasing distance, tapering to zero at the search radius distance from the fault.

We used proximity analysis (Near Distance) to define the distance of each seep from the nearest fault in order to identify the faulting style (normal, strike slip and reverse) of the seep near-field.

Exploratory Spatial Data Analysis allows by charts, graphs, and tables to explore and interpret the correlation among different datasets. In this work, bar charts were used to highlight the spatial relationships between seeps and basins. Histograms and normal probability plots were used to highlight the statistical distribution of the seep distances from the nearest fault system bearing the information of the faulting style (Supplementary Figs. 10 and 11). Histograms were also used to represent and compare fault density data at different scales (global, basins and petroleum fields). Radar diagram was used to investigate the seep distribution in relation to the type of fault.

**Spatial modelling of global geo-CH₄ seepage favourability.** We used a Spatial (GIS-based) MultiCriteria Decision Analysis (SMCDA) to identify the most favourable areas for seepage based on the results of the geospatial analysis, highlighting the relationships among seeps, basins, petroleum fields, heat flow, fault density and fault type (Supplementary Note 5). SMCDA represents a significant evolution of the conventional MCDA in the spatial context[50–53]. We used, in particular, the Analytical Hierarchy Process (AHP) (Supplementary Note 5, Supplementary Tables 4, 5a–b and 6)[54–56].

**Uncertainty of geospatial analysis and original datasets.** Most seeps (2313, 86% of total) have geographic coordinates provided by GPS measurement with uncertainty in the orders of a few metres. For about 14% of seeps (386 seeps) the exact position was not available and geographic coordinates were assigned[57] based on the closest village, generally with the same name of the seep as reported in the literature or other inventories. The actual location of the seep would not exceed, anyway, a few kilometres. Since all inventoried seeps have geographic coordinates with an error <1°, the uncertainty of the spatial location of the seeps at the 1° × 1° grid scale is negligible.

The uncertainty in fault location in the individual datasets used for our global inventory is generally not quantified. The GEM database reports an accuracy corresponding to the denominator of map scale. Since faults were typically compiled and digitised from different maps ranging from 1:100,000 to 1:1,000,000 scale, the accuracy of the fault location should be in the order of 1–10 km. Australia dataset reports a location accuracy between 50 and 1000 m, depending on the quality and scale of the original source data[58]. New Zealand fault database was intended to be represented at 1:250,000 by the designers (GNS Science). Metadata do not report uncertainty and data accuracy[59]. Other used fault databases do not report any indication of uncertainty and accuracy and thus may not have been subject to any verification or other quality control process. Gaps in

the global fault distribution may occur due to the fact that some individual datasets mostly report undefined faults (not included in our integrated dataset), as it happens for Africa, Australia and North America. The final, integrated fault database was checked in terms of geometric duplicates by using the Data Reviewer tool available in ArcGIS. The results of this analysis do not return geometric duplicates.

The first version of the Heat Flow dataset compiled by[60], released by the International Heat Flow Commission[61], indicated that the location of heat flow measurements is of uncertain origin and that the references, as cited in Global Heat Flow Database, were incomplete. GIS capture and quality control were carried out by the Cartographic Services (part of the Geography Department) at Oklahoma State University.

The dataset repository of petroleum fields [https://www.prio.org/Data/Geographical-and-Resource-Datasets/Petroleum-Dataset/Petroleum-Dataset-v-12][62] did not quantify the uncertainty of the petroleum field locations. In the original dataset, several of the sources for field locations were not provided in GIS readable format (e.g., shapefiles), and thus they were digitised from original pdf maps. Details are reported at [https://www.prio.org/Global/upload/CSCW/Data/Geographical/codebook.pdf].

## Data availability

Onshore hydrocarbon seeps. We used the most comprehensive inventory of onshore hydrocarbon seeps, including 2699 gas and oil seeps, and mud volcanoes (MV), derived from ref. [6] (Fig. S1). The dataset includes the seep inventory of CGG[57], which due to license restrictions can be requested at [https://www.cgg.com/en/What-We-Do/Multi-Client-Data/Geological/Robertson-Geochemistry]. Sedimentary Basins. We used the map of the world sedimentary basins from CGG [http://www.datapages.com/associatedwebsites/gisopenfiles/robertstellussedimentarybasinsoftheworldmap.aspx]. We extracted 733 onshore basins classified as convergent and divergent (see Supplementary Note 3). Petroleum fields. We used the global distribution of petroleum (oil and gas) fields from PETRODATA Dataset by PRIO (Peace Research Institute Oslo; [https://www.prio.org/Data/Geographical-and-Resource-Datasets/Petroleum-Dataset/Petroleum-Dataset-v-12/][62]. The dataset includes 891 onshore petroleum fields from 114 countries. It also includes information about the geographic location of hydrocarbon reserves and is specifically designed for display, manipulation and analysis in geographic information systems. Heat Flow Database. International Heat Flow Commission[61]. Global Heat flow database from the International Heat Flow Commission 2011. [http://www.webservice-energy.org/record/3d51419ad85280a84570ef17e880daf89d46be56/]. Fault datasets. A new global dataset of faults was developed combining the GEM (Global Earthquake Model) GAF (Global Active Faults) database and national/regional datasets (listed below). Additional national/regional databases, including non-active faults, available on the web, were also considered. The global fault database includes 114,317 onshore faults (70,732 normal faults; 12,026 strike slip faults; 31,559 reverse faults). We excluded the faults with missing movement information. GEM Global Active Fault Database. The GEM-GAF dataset [https://github.com/GEMScienceTools/gem-global-active-faults] consists of GIS files of shallow fault traces with relevant attributes (i.e., fault geometry, style) and it is licensed under a Creative Commons Attribution License. Global Faults layer from ArcAtlas (ESRI). [http://www.arcgis.com/home/item.html?id=a5496011fa494b99810e4deb5c618ae2#overview]. Afghanistan. U.S. Dep. of the Interior, Data.gov team [https://catalog.data.gov/dataset/geologic-faults-of-afghanistan-fltafg]. Australia. Geoscience Australia and Australian Stratigraphy Commission. (2017). Australian Stratigraphic Units. Bangladesh. U.S. Dep. of the Interior, Data.gov team, [https://catalog.data.gov/dataset/faults-and-tectonic-contacts-of bangladesh-flt8bg]. Caribbean Region. U.S. Dep. of the Interior, Data.gov team [https://catalog.data.gov/dataset/faults-of-the-caribbean-region-flt6bg]. Central Asia. Central Asia Fault Database available at [https://esdynamics.geo.uni-tuebingen.de/faults/]. Crimea. Faults digitised from ref. [63]. Europe. including Turkey. U.S. Dep. of the Interior, Data.gov team [https://catalog.data.gov/dataset/faults-of-europe-including-turkey-flt4-2l]. Georgia. Tectonic map of Georgia from ref. [64]. Greece. [http://diss.rm.ingv.it/share-edsf/]. Iran. U.S. Dep. of the Interior, Data.gov team [https://catalog.data.gov/dataset/major-faults-in-iran-flt2cg]. Ireland. Geological Survey Ireland, Ireland [www.gsi.ie]. Italy. Elementi tettonici presenti nella Carta Geologica d'Italia alla scala 1:100.000. Copyright: Servizio Geologico d'Italia —ISPRA. Portale del Servizio geologico d'Italia [http://sgi.isprambiente.it/geoportal/]. New Zealand. GNS Science, http://data.gns.cri.nz/af/, New Zealand Active fault database. North America. U.S. Geological Survey (and supporting agency if appropriate-see list below), 2006, Quaternary fault and fold database for the United States, accessed DATE, from USGS web site: [https://earthquake.usgs.gov/hazards/qfaults/]. South America. https://github.com/ActiveTectonicsAndes/ATA. Spain. QAFI v.3-Quaternary Active Faults Database of Iberia.: http://info.igme.es/qafi/About.aspx. Switzerland. Opendata Swiss, 2005. Mappa tettonica della Svizzera (GK500-Tekto). Ufficio federale di topografia [http://opendata.swiss/themes/geography]. United Kingdom. British Geological Survey, BGS Geology 625k (DiGMapGB-625) data 1: 625000 ESRI® Faults. [http://www.bgs.ac.uk/products/digitalmaps/dataInfo.html#_625].

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

## Author contributions

G.C. and G.E. conceived the study. G.C. and M.P. carried out the geospatial analysis. G.C., M.P., G.E., U.F. and G.V. analysed and interpreted the data and wrote the manuscript.

## Competing interests

The authors declare no competing interests.
