## [Peer Review File · Nature Communications]

Reviewers' comments:

Reviewer #1 (Remarks to the Author):

Congratulations to you and your coauthors on a well written, timely, and significant research paper on a complex topic.... one that is of interest not only to climate scientists and petroleum explorationists, but also to earth historians more interested in past land-sea-continent configurations and consequences for climate through the ages.

I saw no grammatical or technical errors, but I should note that I did not see reference number 22, Hunt, cited in the text (but may have overlooked it); please check that ...

It was a pleasure to have had the opportunity to review this manuscript.

Dietmar Schumacher

Reviewer #2 (Remarks to the Author):

General points

This paper seeks to investigate the relationship between tectonic setting and global methane emissions using a geospatial analysis approach. The authors compile worldwide datasets for CH₄ seeps, hydrocarbon fields, tectonic setting, faults and heat flow. The authors use a geospatial decision analysis approach that enables a hierarchical consideration of the input parameters to create a global CH₄ favourability model plotted as a map output.

The study is potentially interesting and to my knowledge has not been attempted before so in that sense the work is novel. The authors find that Global geo-CH₄ seems to be preferentially associated with petroleum basins – not really surprising, with a preference for convergent systems – initially slightly surprising until one realises the connection with mud volcanoes which are also prevalent in this setting. The results confirm that geo-CH₄ seems to behave differently from geo-CO₂ on global scales.

However, it's not clear what the main applicability of the results from this particular study is. Will it stimulate more research into microseepage detection, or help with CH₄ budgets for climate models or is it to develop methodologies for methane study? The overall aim and impact could be made more explicit in the abstract, opening sections etc. This vagueness translates into discussion of the limitations of the approach - one obvious limitation is this analysis only considers onshore seeps – and there is no discussion of the effect of not considering the very many large hydrocarbon-rich basins that are located offshore. Can the onshore results be applied offshore? This limitation is less important for method development but more important for global budgets.

The study mainly focuses on Earth but makes some inferences that the same approach could be used on Mars. The problem with this is as it is written, this point just seems out of context – for example - it appears abruptly in the abstract (lines 18 & 19) and there it's not clear why we have gone from discussing CH₄ seepage on Earth to Mars methane. There needs to be some context for this shift in perspective. The same applies to the section at the end of the paper. It appears to be just added on because one of the co-authors has worked on Mars to give a planetary science feel to the paper. There is no discussion of what parts of the model could be applied to Mars and what could not. In this sense, the manuscript doesn't contribute much as a method development paper.

In terms of the input datasets for this analysis, the one I am most concerned, an familiar with, is the global fault data set. Given the absence of such compilations, I was initially impressed that the authors have tried to compile a global fault dataset. The dataset they compiled comes from many sources and some of these are global and others national compilations. These sources are listed in the supplementary datafile. A map of the faults coloured for fault type is given in the supplementary material, however this is very low resolution and its difficult to see the quality of the fault data it contains. There does seem to be some unevenness in the coverage. The fault

mapping feeds into 2 of the parameters in the geospatial analysis – iv) fault density and v) faulting style so it is important to make sure the data compiled are robust. (see comments below).

Specific points

Lines 107 and 113 – it would be good to plot these mean values on Fig 2b as it takes a while to convince yourself these are correct because of all the different data that are included on this plot.

Lines 111-113 – sentence is a bit odd ... 'divergent basins are controlled by tectonic thinning and convergent margins are controlled by tectonic loading'. Surely thinning and loading are the manifestation of the tectonic activity in each case. Stretching in divergent margins produces tectonic thinning and convergence produces tectonic loading? Moreover this sentence seems to be in an odd place as it discusses tectonic setting in the middle of a section on faults and seeps.

Line 174-177 –microseepage is 'the major geo-CH4 source to the atmosphere'. This seems like a pretty important unsupported assertion. If microseepage is so important- who made the estimation, what is the magnitude – how has it been measured? Does it make this studies' results insignificant? How does the authors' model enable an estimation of microseepage? If the main input are seeps not microseeps then how is this possible – its not explained.

Lines 183-185 – this type of model verification is important. Here it is ad-hoc and anecdotal whereas it could be greatly expanded to give a statistical verification. Something like XX% of seeps where structural setting (etc) is known, show agreement with the model.

Line 186 –187. I don't agree that this is intuitive at all given that normal faults are shear displacements. There may be a tendency to produce tensile fractures in their shallowest parts where they may become tensile fractures. These may also exist in the damage zones but normal faults in many (most?) sedimentary basins might be expected to be sealing faults.

Supplementary Files

The supplementary files generally contain a good outline of the method and the underlying data compilations.

Fault compilation comments

Unevenness in coverage. There are comparatively few structures plotted in Africa, for example. In other places the predominant fault type doesn't appear to be correct. I do not see N-S normal faults in Tibet which are commonly developed. The maps here show strike-slip faults instead. The map is of too low a resolution to be able to check these faults properly.

There needs to be more discussion in the supplementary file of :-

Fault resolution –what is the smallest fault resolved – is a uniform cut-off in size applied ? There is mention in the uncertainty section of the GEM map accuracy but nothing mentioned for the national databases that have been used.

Data gaps – where are the main data gaps – such as Africa, Antarctica etc.

Active versus inactive faults – more discussion of active versus inactive faults and their relationship to seep activity. Are seeps mainly found on active structures?

QA and verification: Its not clear how much verification and QA was carried out in areas where the global and regional compilations overlap. For example, I know that in central Italy the GEM model gives a small number of continuous fault traces but regional maps show many more, shorter segmented faults. How has this been dealt with? Has there been a QA process to remove duplicated fault traces that appear in more than one compilation – if not the density values calculated will be meaningless.

The authors need to show in the supplementary data regional maps from their compilation of a few areas such as Turkey, Italy, Tibet, W. US, UK? etc. Alternatively, can the authors make their fault compilation available so that it can be verified against known fault maps?

Ken McCaffrey

December 2019

REPLY TO REVIEWERS

Reviewer #1

I saw no grammatical or technical errors, but I should note that I did not see reference number 22, Hunt, cited in the text (but may have overlooked it); please check that ...

Hunt (1996), as number 22, is cited in the last sentence of section *Seeps and petroleum fields*, when referring to typical heat flows in sedimentary petroleum basins.

Reviewer #2

(...). However, it's not clear what the main **applicability** of the results from this particular study is. Will it stimulate more research into microseepage detection, or help with CH₄ budgets for climate models or is it to develop methodologies for methane study? The overall aim and impact could be made more explicit in the abstract, opening sections etc. This vagueness translates into discussion of the limitations of the approach - one obvious limitation is this analysis only considers onshore seeps – and there is no discussion of the effect of not considering the very many large hydrocarbon-rich basins that are located **offshore**. Can the onshore results be applied offshore? This limitation is less important for method development but more important for global budgets.

The applicability of the results are now better explained in Introduction and we added a sentence in the Abstract. Applicability refers to both atmospheric methane budget (for the better knowledge of the global area where seepage can occur) and petroleum exploration (for better identification of seepage distribution, as indicator of subsurface reservoirs, within petroleum provinces).

We have then added, in the Introduction, a sentence explaining that the work focuses on onshore seepage because detailed offshore seep inventories (with exact geographic location of individual seeps) are not available.

However, the applicability of the results to offshore area is now mentioned at the end of the section "*Seeps and faults*". The exclusion of offshore seeps has not relevance for methane global budget, since this work does not address the emissions but only the relationships between seepage and tectonics. The relative importance of onshore vs offshore seepage in atmospheric methane budget was addressed in other works (Etioppe, G., Ciotoli, G., Schwietzke, S., Schoell, M., *Gridded maps of geological methane emissions and their isotopic signature. Earth Syst. Sci. Data*, **11**, 1–22, (2019). Etioppe, G., Schwietzke, S., *Global geological methane emissions: an update of top-down and bottom-up estimates. Elem. Sci. Anth.* **7**, 47, (2019)).

The study mainly focuses on Earth but makes some inferences that the same approach could be used on **Mars**. The problem with this is as it is written, this point just seems out of context – for example - it appears abruptly in the abstract (lines 18 & 19) and there it's not clear why we have we gone from discussing CH₄ seepage on Earth to Mars methane. There needs to be some context for this shift in perspective. The same applies to the section at the end of the paper. It appears to be just added on because one of the co-authors has worked on Mars to give a planetary science feel to the paper. There is no discussion of what parts of the model could be applied to Mars and what could not. In this sense, the manuscript doesn't contribute much as a method development paper.

We agree that the reference to Mars appears abruptly in the Abstract. With the addition of the sentence on the applicability (previously suggested by the Reviewer) the reference to Mars is now contextualized.

Concerning the section at the end of the paper, we believe to have clearly explained why and how the results of the study can be useful for Mars. It is not the seepage favourability model that can be applied to Mars, since the model is based on geological parameters that are unknown on Mars (basin type – convergent or divergent, petroleum fields; heat flow), but just the seepage vs tectonics spatial relationship. It can be stated, by analogy with what is observed on Earth, that any type of fault on Mars, regardless the fault density of the area, may potentially be methane-bearing. We believe this concept can be highly appreciated by the Mars scientific community.

In terms of the input datasets for this analysis, the one I am most concerned, and familiar with, is the global **fault data set**. Given the absence of such compilations, I was initially impressed that the authors have tried to compile a global fault dataset. The dataset they compiled comes from many sources and some of these are global and others national compilations. These sources are listed in the supplementary datafile. A map of the faults coloured for fault type is given in the supplementary material, however this is very low resolution and it is difficult to see the quality of the fault data it contains. There does seem to be some unevenness in the coverage. The fault mapping feeds into 2 of the parameters in the geospatial analysis – iv) fault density and v) faulting style so it is important to make sure the data compiled are robust. (see comments below).

Clarifications on data coverage, duplications and accuracy are reported below, answering to specific questions

Specific points

Lines 107 and 113 – it would be good to plot these mean values on Fig 2b as it takes a while to convince yourself these are correct because of all the different data that are included on this plot.

We agree, and we have added the mean values as suggested.

Lines 111-113 – sentence is a bit odd ... 'divergent basins are controlled by tectonic thinning and convergent margins are controlled by tectonic loading'. Surely thinning and loading are the manifestation of the tectonic activity in each case. Stretching in divergent margins produces tectonic thinning and convergence produces tectonic loading? Moreover this sentence seems to be in an odd place as it discusses tectonic setting in the middle of a section on faults and seeps.

We agree and removed the sentence as it is not relevant in the context of the section. The concept was already reported in the section "Seeps and basin geodynamics".

Line 174-177 –microseepage is 'the major geo-CH₄ source to the atmosphere'. This seems like a pretty important unsupported assertion. If microseepage is so important- who made the estimation, what is the magnitude – how has it been measured? Does it make this studies' results insignificant? How does the authors' model enable an estimation of microseepage? If the main input are seeps not microseeps then how is this possible – its not explained.

We have now added reference and magnitude for microseepage emission, showing the importance of the applicability of our results to the knowledge of global microseepage. We have also explained that the favourability model is applicable to any type of seepage, focused (seeps or macro-seeps) and diffused (microseepage), since the gas migration mechanism (fundamentally advective, i.e., driven by pressure gradients, and whose intensity is mainly controlled by fault-related permeability) is the same.

Lines 183-185 – this type of model verification is important. Here it is ad-hoc and anecdotal whereas it could be greatly expanded to give a statistical verification. Something like XX% of seeps where structural setting (etc) is known, show agreement with the model.

We understand and agree in principle, but unfortunately the documented and well constrained examples of correlation between fault type and seep occurrence are not enough for a global statistical analysis. We provided the best examples (five references) as reported in the literature.

In the Introduction we stated that “...but the role of tectonics in degassing is known only for a few case studies at local scale.”

Line 186 –187. I don't agree that this is intuitive at all given that normal faults are shear displacements. There may be a tendency to produce tensile fractures in their shallowest parts where they may become tensile fractures. These may also exist in the damage zones but normal faults in many (most?) sedimentary basins might be expected to be sealing faults.

We agree and have deleted the sentence. We only wanted to outline that scholars, not expert on structural geology and seepage, could erroneously think that normal faults are more permeable than reverse faults. Some of us assisted in discussions on this theme, where several scholars argued that normal faults may more easily host gas seepage. We then clarify, as evidenced by our analysis, that this is not true.

Supplementary Files

The supplementary files generally contain a good outline of the method and the underlying data compilations.

Fault compilation comments

Unevenness in coverage. There are comparatively few structures plotted in Africa, for example. In other places the predominant fault type doesn't appear to be correct. I do not see N-S normal faults in Tibet which are commonly developed. The maps here show strike-slip faults instead. The map is of too low a resolution to be able to check these faults properly.

There needs to be more discussion in the supplementary file of :-

All faults from the considered databases with defined kinematics (normal, strike slip and reverse) are plotted. We highlight that the undefined faults are not considered for the analysis and then are not included. Africa, Australia and North America databases reported mostly undefined faults and this is the reason why they show major fault gaps in our final database. More detailed explanations on data gaps are now reported in the Supplementary Information (Section S4).

N-S normal faults in Tibet are included in the considered global dataset; unfortunately, they were not visible in the map of the Supplementary Information (Fig.S6) because, during the layout exporting process into ArcGIS, a previous layer, relative to a local fault database, with not-checked data was erroneously turned on. We have corrected this issue and provide a new Fig.S6 with the right checked layers.

In this respect, we would like to remark that all considered fault databases are re-checked in terms of duplicates by using the Data Reviewer tool available in ArcGIS. The results of this analysis did not return duplicates. We have added a short description in the Supplementary Information.

Fault resolution –what is the smallest fault resolved – is a uniform cut-off in size applied? There is mention in the uncertainty section of the GEM map accuracy but nothing mentioned for the national databases that have been used.

Unfortunately, regional datasets never report fault resolution and accuracy. For example, regarding the Afghanistan fault dataset, the USGS reports that “*although all data and software published are used by the USGS, no warranty, expressed or implied, is made by the USGS as to the accuracy of the data and related materials*”.

More detailed explanations on fault resolution are now reported in the Supplementary Information (Section S4).

Data gaps – where are the main data gaps – such as Africa, Antarctica etc.

Global coverage is commented above. The fault and seep datasets we used do not report data on Antarctica.

Active versus inactive faults – more discussion of active versus inactive faults and their relationship to seep activity. Are seeps mainly found on active structures?

GEM includes only active faults, the other datasets do not distinguish active from inactive faults, so the role of fault activity cannot be addressed in our geospatial analysis. This is now explained in the text (end of section *Consistency with local scale observations...*). We however reported, already in the previous version, that gas migration studies demonstrated that fault activity is not relevant for gas seepage, as inactive, ancient faults in stable and poorly fractured terrains (e.g., in granites, within cratons) can also be gas-bearing (ref 43-44).

QA and verification: Its not clear how much verification and QA was carried out in areas where the global and regional compilations overlap. For example, I know that in central Italy the GEM model gives a small number of continuous fault traces but regional maps show many more, shorter segmented faults. How has this been dealt with? Has there been a QA process to remove duplicated fault traces that appear in more than one compilation – if not the density values calculated will be meaningless.

Yes, duplicated elements (faults) were checked at the beginning using Data Reviewer tool available in ArcGIS. We initially found n.930 duplicates between GEM database and the European Database of Seismogenic Faults. These duplicates were deleted and not integrated in our final database. As indicated above, the duplication test has been repeated in this revision, and no more duplicates have been found.

The authors need to show in the supplementary data regional maps from their compilation of a few areas such as Turkey, Italy, Tibet, W. US, UK? etc. Alternatively, can the authors make their fault compilation available so that it can be verified against known fault maps?

OK, we have added in Supplementary some regional maps as requested (Fig. S7a-e).

REVIEWERS' COMMENTS:

Reviewer #2 (Remarks to the Author):

The revisions have helped to clarify and strengthen the claims of this study further and I now consider that it deserves to be published in Nature Communications. Most of my concerns have been allayed. By adding further clarification in the supplementary the limits of the study are clearer.